# Circulating Cell-Free mtDNA Contributes to AIM2 Inflammasome-Mediated Chronic Inflammation in Patients with Type 2 Diabetes

**DOI:** 10.3390/cells8040328

**Published:** 2019-04-08

**Authors:** Jung Hwan Bae, Seung Jo, Seong Jin Kim, Jong Min Lee, Ji Hun Jeong, Jeong Suk Kang, Nam-Jun Cho, Sang Soo Kim, Eun Young Lee, Jong-Seok Moon

**Affiliations:** 1Department of Integrated Biomedical Science, Soonchunhyang Institute of Medi-bio Science (SIMS), Soonchunhyang University, Cheonan-si, Chungcheongnam-do 31151, Korea; junier24@naver.com (J.H.B.); 309cho@naver.com (S.J.II); tjdwls515@naver.com (S.J.K.); whdals7034@naver.com (J.M.L.); jihun@sch.ac.kr (J.H.J.); 2Department of Internal Medicine, Soonchunhyang University Cheonan Hospital, Cheonan 31151, Korea; abalonea@naver.com (J.S.K.); c100086@schmc.ac.kr (N.-J.C.); 3Institute of Tissue Regeneration, College of Medicine, Soonchunhyang University, Cheonan 31151, Korea; 4Department of Internal Medicine, Pusan National University Hospital, Pusan 50612, Korea; drsskim7@gmail.com

**Keywords:** ccf-mtDNA, chronic inflammation, AIM2 inflammasome, type 2 diabetes

## Abstract

Mitochondrial dysfunction has been implicated in the pathogenesis of insulin resistance and type 2 diabetes. Damaged mitochondria DNA (mtDNA) may have a role in regulating hyperglycemia during type 2 diabetes. Circulating cell-free mitochondria DNA (ccf-mtDNA) was found in serum and plasma from patients and has been linked to the prognosis factors in various human diseases. However, the role of ccf-mtDNA in chronic inflammation in type 2 diabetes is unclear. In this study, we hypothesized that the ccf-mtDNA levels are associated with chronic inflammation in patients with type 2 diabetes. The mtDNA levels were elevated in the plasma from patients with type 2 diabetes compared to healthy subjects. The elevated mtDNA levels were associated with interleukin-1β (IL-1β) levels in patients with type 2 diabetes. The mtDNA, from patients with type 2 diabetes, induced absent in melanoma 2 (AIM2) inflammasome-dependent caspase-1 activation and IL-1β and IL-18 secretion in macrophages. Our results suggest that the ccf-mtDNA might contribute to AIM2 inflammasome-mediated chronic inflammation in type 2 diabetes.

## 1. Introduction

Chronic inflammation is linked to insulin resistance and impaired glucose homeostasis in type 2 diabetes [1]. Inflammasomes are multi-protein complexes that activate caspase-1 and downstream immune responses, including the maturation and secretion of interleukin-1β (IL-1β) and IL-18 in immune cells such as macrophages [2,3]. Interleukin-1β is one of key pro-inflammatory cytokines during chronic inflammation in type 2 diabetes [4]. IL-1β interferes with insulin signaling in insulin-sensitive cells [5]. We have previously reported that cellular metabolic pathways, which are related to type 2 diabetes, are associated with the nucleotide-binding domain leucine-rich repeat containing receptor (NLR) and the pyrin domain containing 3 (NLRP3) inflammasome activation in macrophages under pro-inflammatory conditions [6,7].

Absent in melanoma 2 (AIM2) inflammasomes sense double strand DNA (dsDNA), and they interact with the adaptor apoptosis-associated speck-like protein containing a caspase recruitment domain (ASC), which recruits pro-caspase-1 that regulates an immune response, by activation of caspase-1, and the maturation and secretion of IL-1β and IL-18 [8,9]. AIM2 inflammasome contributes to the development of human diseases including psoriasis, dermatitis, arthritis, and inflammatory diseases [10]. A previous study suggests that AIM2 inflammasome expression and activation is increased, and may affect systemic inflammation and organ failure in patients with an early course of acute pancreatitis [11]. In addition, the AIM2 expression was found in kidneys from patients with diabetic or nondiabetic chronic kidney disease (CKD) [12].

Mitochondrial damage associated molecular pattern molecules (DAMPs), such as mitochondrial DNA (mtDNA), ATP, and cytochrome-c can be released from damaged mitochondria in various death cells or injured tissues as endogenous danger signals [13,14,15,16,17,18]. The release of mitochondrial DAMPs into the circulation is linked to systemic inflammatory responses [19]. The mitochondrial DAMPs are associated with the inflammatory response in human diseases [20,21,22,23]. Among the mitochondrial DAMPs, circulating cell-free mitochondrial DNA (ccf-mtDNA) in plasma has been found in various human diseases [24,25,26,27,28,29]. Previous studies suggest that the alterations of ccf-mtDNA levels correlated with progressive cellular stress and death in various chronic diseases such as diabetes, coronary heart diseases, Parkinson’s disease, and Alzheimer’s disease [24,25,26,27,28,29]. A recent study has shown that ccf-mtDNA was elevated in type 2 diabetes patients with coronary heart disease (CHD) and correlated with C-reactive protein (CRP) levels [25]. Currently, the role of ccf-mtDNA in AIM2 inflammasome activation-dependent chronic inflammation during type 2 diabetes remains unclear.

In this study, we demonstrate that ccf-mtDNA contributes to AIM2 inflammasome-mediated chronic inflammation in patients with type 2 diabetes. Our results show that the ccf-mtDNA levels were elevated in plasma from patients with type 2 diabetes. In addition, our results show that the IL-1β levels were increased in plasma from patients with type 2 diabetes. The mtDNA from patients with type 2 diabetes induced AIM2 inflammasome-dependent caspase-1 activation, and IL-1β and IL-18 secretion in macrophages. Our results, taken together, suggest that the ccf-mtDNA-mediated AIM2 inflammasome activation provides a mechanism for chronic inflammation in type 2 diabetes.

## 2. Materials and Methods

### 2.1. Human Subjects Study

Human subjects study was conducted in accordance with the Helsinki Declaration. The protocol was approved by the Institutional Review Board of Pusan National University Hospital (20100024). All patients provided written informed consent before enrollment. The subjects were collected from a prospective observational study for diabetic nephropathy (Diabetic Kidney Disease Study [DKDS]) [30,31,32]. A total of 172 patients including 147 type 2 diabetic patients and 25 healthy controls were enrolled consecutively between February 2010 and February 2012 at the outpatient clinics. The eligibility criteria (inclusion and exclusion) has been described in previous studies [31,32]. Briefly, all enrolled patients had relatively conserved renal function, their estimated glomerular filtration rates (eGFR) were ≥60 mL/min/1.73 m^2^, and serum creatinine was <1.2 mg/dL. In addition, patients had a sufficient washout period for renin–angiotensin system (RAS) inhibitors. Patients had no history of administration of RAS inhibitors at enrollment, or they had a washout period of at least 2 months for these drugs before enrollment. After enrollment, both medical histories and anthropometric measurements were obtained at the initial clinic visit. Blood samples were obtained at the same visit. Among the 172 patients, 9 were excluded because of the lack of baseline plasma or spot urine sample. The remaining 163 patients included in this study consisted of 141 type 2 diabetic patients and 22 healthy controls. A large study was difficult in local public health agencies and communities. The results were produced with limited available resources.

### 2.2. Reagents and Antibodies

Ultrapure lipopolysaccharide (LPS) (tlrl-3pelps), flagellin (*Salmonella typhimurium*) (tlrl-stfla), and MDP (tlrl-mdp) were from Invivogen (San Diego, CA, USA). The poly(dA:dT) (P0883) and ATP (A2383) were from Sigma-Aldrich (St. Louis, MO, USA). The Z-VAD-FMK (2163) was from Tocris Bioscience (Minneapolis, MN, USA). The following antibodies were used: polyclonal rabbit anti-caspase-1 for mouse caspase-1 (1:1000) (SC-514, Santa Cruz Biotechnology, Dallas, TX, USA); polyclonal goat anti-IL-1β for mouse IL-1β (1:1000) (AF-401-NA, R&D Systems, Minneapolis, MN, USA); polyclonal rabbit anti-ASC for mouse ASC (1:1000) (ADI-905-173-100, Enzo Life Sciences, Farmingdale, NY, USA); and monoclonal mouse anti-β-actin (1:5000) (A5316, Sigma-Aldrich).

### 2.3. Isolation of ccf-DNA from Plasma in Human Subjects

Plasma was collected in EDTA-coated blood collection tubes. The plasma samples were stored at −80 °C. The ccf-DNA was isolated from the plasma (100 μL) using a Maxwell^®^ RSC Instrument (AS4500, Promega, Madison, WI, USA) and a Maxwell^®^ RSC ccf-DNA Plasma Kit (AS1480, Promega), according to the manufacture’s manual. At the final elution step, isolated ccf-DNA was eluted using 50 μL 1 × TE buffer according to the manufacture’s manual.

### 2.4. Quantification of ccf-DNA Levels from Plasma in Human Subjects

The quantitation of dsDNA and single strand DNA (ssDNA) concentrations in ccf-DNA was measured by fluorescent detection using a Quantus™ Fluorometer (E6150, Promega), according to the manufacture’s manual. The concentrations of dsDNA were determined using QuantiFluor^®^ dsDNA System (E2670, Promega). The concentrations of ssDNA were determined using QuantiFluor^®^ ssDNA System (E3190, Promega).

### 2.5. Quantification of mtDNA and nDNA Levels in ccf-DNA from Plasma in Human Subjects

For the standard of mtDNA copy number, human nicotinamide adenine dinucleotide (reduced) (NADH) dehydrogenase 1 (MTND1) cDNA clone (SC101172, OriGene Technologies, Rockville, MD, USA) was used. For the standard of nuclear DNA (nDNA) copy number, human hemoglobin, beta (HBB) cDNA ORF clone (Myc-DDKtagged) (RC203258, OriGene Technologies) was used. Concentrations were converted to copy number using the formula, mol/gram × molecules/mol = molecules/gram, via a DNA copy number calculator [33], University of Rhode Island Genomics and Sequencing Center, Kingston, RI, USA) [19,34]. A 10 μL mixture containing 5 μL of ccf-DNA and a set of gene-specific primers was mixed with 10 μL of 2 × SYBR Green PCR Master Mix (4309155, Applied Biosystems, Waltham, MA, USA) and then subjected to Realtime-polymerase chain reaction (RT-PCR) quantification using the ABI PRISM 7300 real-time PCR system (Applied Biosystems). The primer sequences were as follows: human NADH dehydrogenase 1 gene (mtDNA), forward 5′-ATACCCATGGCCAACCTCCT-3′, reverse 5′-GGGCCTTTGCGTAGTTGTAT-3′ [19]; human β-globin (nuclear DNA), forward 5′-GTGCACCTGACTCCTGAGGAGA-3′, reverse 5′-CCTTGATACCAACCTGCCCAG-3′ [19]. The RT-PCR assay for detecting mtDNA was carried out as follows: Initiated by a denaturation step for 10 min at 95 °C; and a further step consisting of 50 cycles for 30 s at 95 °C, 30 s at 60 °C, and 30 s at 72 °C. The levels of mtDNA in all plasma analyses were expressed in copies per microliter of plasma based on the following calculation [19,34,35]:

C = Q × VDNA/VPCR × 1/VEXT
(1)
where C is the level of DNA in plasma (copies/microliter plasma); Q is the quantity (copies) of DNA determined by the sequence detector in a PCR; VDNA is the total volume of plasma ccf-DNA solution obtained after isolation, 50 μL; VPCR is the volume of plasma ccf-DNA solution used for PCR, 5 μL; and VEXT is the volume of plasma extracted, 100 μL.

### 2.6. Cell Culture

All mouse experimental protocols were approved by the Institutional Animal Care and Use Committee of Soonchunhyang University (protocol #: SCH18-0032). For primary mouse bone marrow-derived macrophages (BMDMs), bone marrow collected from wild type (WT) mouse (male, 8–10 weeks old) femurs and tibias was plated on sterile Petri dishes and incubated for 7 d in a Dulbecco’s modified Eagle’s medium (DMEM) (Invitrogen, Waltham, MA, USA) containing 10% (*v*/*v*) heat-inactivated fetal bovine serum (FBS), 100 units/mL penicillin, 100 mg/mL streptomycin, and 25% (*v*/*v*) conditioned medium from mouse L929 fibroblasts (CCL-1, ATCC, Manassas, VA, USA). For the AIM2 inflammasome activation, LPS-primed wild-type bone marrow-derived macrophages (BMDMs) were transfected with poly(dA:dT) (1 mg/mL, 3 h) (Sigma-Aldrich) using Lipofectamine with Plus reagent (15338-100, Invitrogen), according to the manufacturer’s instructions. For the NLRP3 inflammasome activation, LPS-primed WT BMDMs were treated with ATP (2 mM, 0.5 h) (Sigma-Aldrich). For the NLRC4 inflammasome activation, LPS-primed WT BMDMs were transfected with flagellin (5 μg/mL, 6 h) (Invivogen) using Lipofectamine with Plus reagent (15338-100, Invitrogen), according to the manufacturer’s instructions. For the NLRP1 inflammasome activation, LPS-primed WT BMDMs were treated with MDP (5 μg/mL, 16 h) (Invivogen). For cytokine analysis, the cell supernatants and cell lysates were collected and analyzed for the levels of IL-1β, IL-18, and TNF-α using a ELISA kit.

### 2.7. Immunoblot Analysis

The WT BMDMs were harvested and lysed in NP40 Cell Lysis Buffer (FNN0021, Invitrogen). The lysates were centrifuged at 15,300× *g* for 10 min at 4 °C, and the supernatants were obtained. The protein concentrations of the supernatants were determined by applying the Bradford assay (500-0006, Bio-Rad Laboratories, Hercules, CA, USA). Proteins were electrophoresed on NuPAGE 4−12% Bis-Tris gels (Invitrogen) and transferred to Protran nitrocellulose membranes (10600001, GE Healthcare Life Science, Pittsburgh, PA, USA). The membranes were blocked in 5% (*w*/*v*) bovine serum albumin (BSA) (9048-46-8, Santa Cruz Biotechnology) in TBS-T (TBS (170-6435, Bio-Rad Laboratories) and 1% (*v*/*v*) Tween-20 (170-6531, Bio-Rad Laboratories)) for 30 min at 25 °C. The membranes were incubated with primary antibody diluted in 1% (*w*/*v*) BSA in TBS-T for 16 h at 4 °C and then with the horseradish peroxidase (HRP) conjugated secondary antibody (anti-rabbit IgG-HRP (SC-2357, Santa Cruz Biotechnology) (1:2500) and anti-mouse m-IgGκ BP-HRP (SC-516102, Santa Cruz Biotechnology) (1:2500)) diluted in TBS-T for 30 min at 25 °C. The immunoreactive bands were detected using the SuperSignal West Pico Chemiluminescent Substrate (34078, Thermo Scientific, Waltham, MA, USA).

### 2.8. Cytokine Analysis

Supernatants from BMDMs were measured for mouse IL-1β (MLB00C, R&D systems), mouse IL-18 (7625, R&D systems), mouse TNF-α (MTA00B, R&D systems) according to the manufacturer’s instructions. Plasma in human subjects was measured for human IL-1β (DLB50, R&D systems) according to the manufacturer’s instructions.

### 2.9. ASC Oligomerization and ASC Speck Formation

The WT BMDMs were seeded on chamber slides. After LPS and poly(dA:dT) stimulation, cells were fixed with 4% paraformaldehyde and then incubated with polyclonal ASC antibody (ADI-905-173-100, Enzo Life Sciences) for 16 h and FITC goat anti-rabbit (IgG) secondary antibody (ab6717, Abcam, Cambridge, MA, USA) for 1h followed by DAPI (P36962, Thermo Fisher Scientific) staining [7,36,37]. The ASC specks were analyzed using a Zeiss LSM880 laser scanning confocal microscope and quantified using ImageJ software v1.52a (NIH, Bethesda, MD, USA). The graph in Figure 4D represents the quantification of percent of ASC speck positive cells for each mouse.

### 2.10. Statistical Analysis

All data are mean ± SD, combined from three independent experiments. All statistical tests were analyzed using a two-tailed Student’s *t*-test for comparison of two groups, and analysis of variance (ANOVA) (with post hoc comparisons using Dunnett’s test), using a statistical software package (GraphPad Prism version 4.0, GraphPad Software Inc., San Diego, CA, USA) for comparison of multiple groups. The correlation tests were analyzed using the Spearman correlation coefficient (Spearman’s R) for comparison of two groups. The strength of difference was analyzed using the Cohen’s d effect size (d) for comparison of two groups. The linear regression analysis (r^2^) was analyzed in comparison of two groups, and *p*-values less than 0.05 were considered statistically significant.

## 3. Results

### 3.1. The ccf-DNA Levels Were Increased in Plasma from Patients with Type 2 Diabetes

To investigate the role of ccf-mtDNA in type 2 diabetes, we first analyzed the whole ccf-DNA levels in the plasma from patients with type 2 diabetes. We isolated the whole ccf-DNA including single strand DNA (ssDNA) and double strand DNA (dsDNA) in the plasma from patients with type 2 diabetes. The ccf-DNA was isolated from the plasma of 141 patients with type 2 diabetes and 22 healthy subjects without type 2 diabetes as controls (Table 1). After isolation of ccf-DNA from the plasma, we measured the DNA levels through the quantification of ssDNA and dsDNA concentration in ccf-DNA from the plasma. The concentrations of both ssDNA and dsDNA were higher in the plasma from patients with type 2 diabetes (T2D) relative to that in healthy subjects (control) (Figure 1A,B). These results suggest that the ccf-DNA levels were increased in plasma from patients with type 2 diabetes.

### 3.2. The mtDNA Levels Were Elevated in Plasma from Patients with Type 2 Diabetes

Next, we investigated whether the mtDNA levels were elevated in patients with type 2 diabetes. We analyzed the mtDNA levels in the plasma from 141 patients with type 2 diabetes and 22 healthy subjects without type 2 diabetes as controls. We measured the mtDNA levels through the quantification of copy number for mtDNA in ccf-DNA from the plasma. Notably, the mtDNA levels were significantly higher in patients with type 2 diabetes (T2D) compared to healthy subjects (control) (Figure 2). We examined the elevated mtDNA levels to determine whether they correlated with HbA1c, which is an indicator of high blood glucose levels [38,39], for the diagnosis of type 2 diabetes. There was a weak correlation between the elevated mtDNA levels and HbA1c levels in the plasma from patients with type 2 diabetes (Appendix A). In contrast, the nDNA levels were lower than the mtDNA levels in plasma from patients with type 2 diabetes (Appendix A). These results suggest that the mtDNA levels were elevated in patients with type 2 diabetes.

### 3.3. The IL-1β Levels Were Elevated in Plasma from Patients with Type 2 Diabetes

To investigate the role of mtDNA in chronic inflammation in type 2 diabetes, we analyzed whether the elevated mtDNA levels could affect the IL-1β levels, which is a critical pro-inflammatory cytokine regulated by inflammasomes during chronic inflammation, in patients with type 2 diabetes. We measured the IL-1β levels in plasma from 141 patients with type 2 diabetes and 22 healthy subjects without type 2 diabetes as controls. Notably, the IL-1β levels were significantly higher in the plasma from patients with type 2 diabetes (T2D) compared to healthy subjects (control) (Figure 3). Next, we examined the elevated mtDNA levels to determine whether they correlated with the IL-1β levels in the plasma from patients with type 2 diabetes. There was a weak correlation between the elevated mtDNA levels and IL-1β levels in plasma from patients with type 2 diabetes (Appendix A). These results suggest that the elevated mtDNA levels were associated with IL-1β-mediated chronic inflammation in type 2 diabetes.

### 3.4. The mtDNA from Patients with Type 2 Diabetes Induced AIM2 Inflammasome Activation in Macrophages

To investigate a molecular target of ccf-mtDNA in regulating chronic inflammation in type 2 diabetes, we examined ccf-DNA, which has high mtDNA levels, from patients with type 2 diabetes to determine whether it could induce AIM2 inflammasome activation in macrophages. We analyzed caspase-1 activation and the IL-1β and IL-18 secretion in lipopolysaccharide (LPS)-primed bone marrow-derived macrophages (BMDMs) stimulated with ccf-DNA from patients with type 2 diabetes and poly(dA:dT), an AIM2 inflammasome activator. Notably, the ccf-DNA (ccf-DNA #1 and ccf-DNA #2) from two independent patients with type 2 diabetes resulted in a higher expression of cleaved caspase-1 and cleaved IL-1β in response to poly(dA:dT) stimulation compared with the vehicle control (control), although pro-IL-1β expression was unchanged (Figure 4A). Consistently, the ccf-DNA (ccf-DNA #1 and ccf-DNA #2) from two independent patients with type 2 diabetes showed significantly higher IL-1β and IL-18 secretion in response to poly(dA:dT) as compared with the vehicle control (Figure 4B), whereas the secretion of tumor necrosis factor (TNF)-α was unchanged (Figure 4B). Moreover, we examined whether mtDNA-induced AIM2 inflammasome activation is critical for caspase-1 activation-dependent IL-1β and IL-18 secretion. We analyzed caspase-1 activation and IL-1β and IL-18 secretion in BMDMs pre-treated with Z-VAD, a selective caspase-1 inhibitor, before poly(dA:dT) and ccf-DNA stimulation after LPS incubation. Z-VAD suppressed caspase-1 activation, IL-1β cleavage, and secretion of IL-1β and IL-18 in response to ccf-DNA and poly(dA:dT) stimulation relative to the vehicle control, while TNF-α was unchanged (Appendix A). These results suggest that ccf-DNA-induced AIM2 inflammasome activation is required for caspase-1-dependent IL-1β and IL-18 secretion.

In contrast, the ccf-DNA (ccf-DNA #1 and ccf-DNA #2) had no effect on the secretion of IL-1β and IL-18 in response to ATP (a NLRP3 inflammasome activator), or flagellin, (a NLRC4 inflammasome activator), or muramyldipeptide (MDP), (a NLRP1 inflammasome activator) as shown in Figure 4C. Furthermore, we investigated whether the ccf-DNA from type 2 diabetes promoted the formation of ASC specks, which is required for NLRP3-dependent caspase-1 activation [7]. The ccf-DNA from patients with type 2 diabetes showed significantly higher formation of ASC specks that were induced by LPS and poly(dA:dT) stimulation as compared with the vehicle control (Figure 4D). These results suggest that the mtDNA from patients with type 2 diabetes induces AIM2 inflammasome activation in macrophages.

## 4. Discussion

In this study, we demonstrate that elevated ccf-mtDNA levels contribute to chronic inflammation via AIM2 inflammasome activation in patients with type 2 diabetes. Our results show that the mtDNA levels were elevated in the plasma from patients with type 2 diabetes. Moreover, we show that the IL-1β levels were elevated in the plasma from patients with type 2 diabetes. Furthermore, the mtDNA from patients with type 2 diabetes increased AIM2 inflammasome activation in macrophages.

The elevated ccf-mtDNA levels are linked to the risk and outcome prediction in various human diseases including myocardial infarction, trauma, sepsis, malignancy, and intensive care unit conditions [10,35,40,41,42,43]. Consistent with previous studies, our results show that ccf-mtDNA levels are elevated in patients with type 2 diabetes. In addition, we found that the ccf-mtDNA levels are significantly higher than ccf-nDNA in the plasma in patients with type 2 diabetes. Our results suggest that mitochondria may be a major source of ccf-DNA in patients with type 2 diabetes.

Since mitochondrial dysfunction is associated with insulin resistance and type 2 diabetes [44], dysfunctional mitochondria can trigger release of mtDNA into extracellular space during cellular injury and death. Chronic hyperglycemia in patients with type 2 diabetes causes the production of reactive oxygen species (ROS) which causes oxidative damage and mitochondrial dysfunction [45]. Our results show that the HbA1c levels are associated with the elevated mtDNA levels in the plasma from patients with type 2 diabetes. Although our results show the interaction between the elevated mtDNA and HbA1c levels, the correlation of these two factors is weak. These results suggest that other factors could be involved in the elevated mtDNA levels in patients with type 2 diabetes. Therefore, the regulation of ccf-mtDNA levels in type 2 diabetes needs to be clarified by further studies.

Chronic inflammation during type 2 diabetes is linked to inflammasome activation. Previous studies suggest that high-fat diet (HFD) or free fatty acids (FFAs) increase NLRP3 inflammasome activation in adipose tissues and macrophages [5,46,47]. Although the role of NLRP3 inflammasome activation has been demonstrated, the activation of AIM2 inflammasome during type 2 diabetes is not well known. Our results suggest that ccf-mtDNA-dependent AIM2 inflammasome activation is associated with chronic inflammation in patients with type 2 diabetes. Moreover, our results suggest that ccf-mtDNA regulates IL-1β levels in patients with type 2 diabetes. Consistent with our findings, ccf-mtDNA is linked to chronic inflammation in patients with hemodialysis and cardiovascular diseases [48,49]. Our findings suggest that the ccf-mtDNA may act as a critical signaling molecule in chronic inflammation via AIM2 inflammasome activation in metabolic diseases, including type 2 diabetes. Although we suggest that the elevated ccf-mtDNA levels contribute to IL-1β-dependent chronic inflammation in patients with type 2 diabetes, our results showed very weak correlations between elevated ccf-mtDNA and IL-1β levels with a small sample size for healthy subjects as a control in comparison with patients with type 2 diabetes. Further work will be needed to overcome these limitations.

In summary, we found that the elevated ccf-mtDNA levels may contribute to chronic inflammation in patients with type 2 diabetes. The ccf-mtDNA levels were increased in patients with type 2 diabetes. The ccf-mtDNA promotes AIM2 inflammasome activation in macrophages. Given the relationship between the circulating mtDNA and chronic inflammation, our findings may have broad implications for the understanding of chronic inflammation in type 2 diabetes.

## Figures and Tables

**Figure 1 cells-08-00328-f001:**
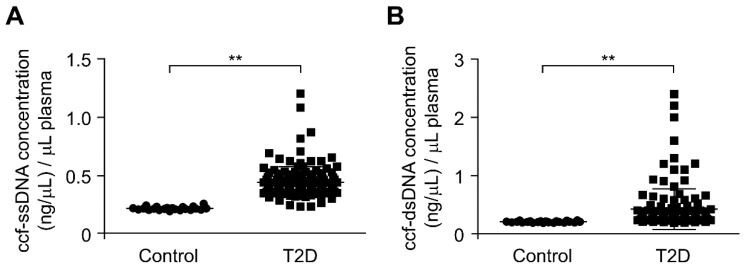
The circulating cell-free mitochondria (ccf-DNA) levels were increased in plasma from patients with type 2 diabetes. (**A**) Quantification of single strand DNA (ssDNA) concentration in ccf-DNA (ccf-ssDNA) from plasma of 22 healthy subjects (control) and 141 patients with type 2 diabetes (T2D). Data are mean ± SEM. ** *p* < 0.01 using the two-tailed Student’s *t*-test. Cohen’s d effect size *d* = 1.84. (**B**) Quantification of double strand DNA (dsDNA) concentration in ccf-DNA (ccf-dsDNA) from plasma of 22 healthy subjects (control) and 141 patients with type 2 diabetes (T2D). Data are mean ± SEM. ** *p* < 0.01 by two-tailed Student’s *t*-test. Cohen’s d effect size *d* = 0.69.

**Figure 2 cells-08-00328-f002:**
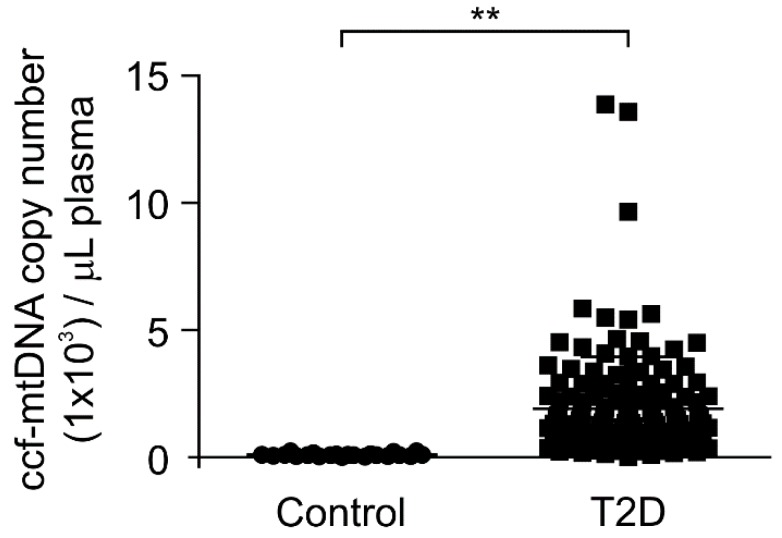
The mitochondria DNA (mtDNA) levels were elevated in plasma from patients with type 2 diabetes. Quantitative PCR analysis for mtDNA levels in ccf-DNA (ccf-mtDNA) from plasma of 22 healthy subjects (control) and 141 patients with type 2 diabetes (T2D). Data are mean ± SEM. ** *p* < 0.01 using the two-tailed Student’s *t*-test.

**Figure 3 cells-08-00328-f003:**
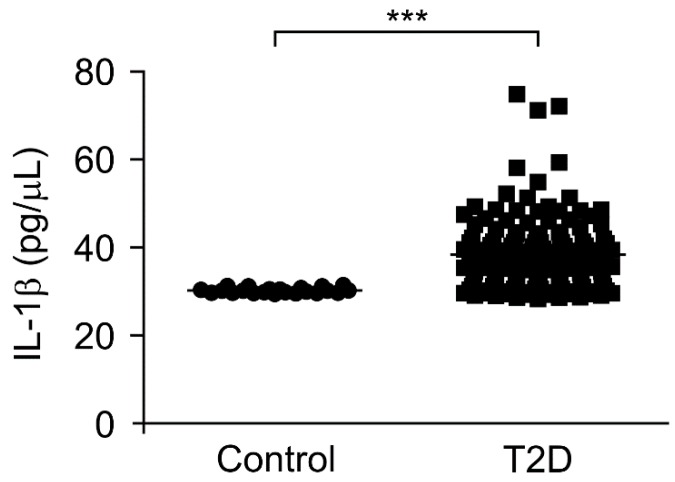
The interleukin-1β (IL-1β) levels were elevated in plasma from patients with type 2 diabetes. Quantification of IL-1β levels from plasma of 22 healthy subjects (control) and 141 patients with type 2 diabetes (T2D). Data are mean ± SEM. *** *p* < 0.001 using the two-tailed Student’s *t*-test.

**Figure 4 cells-08-00328-f004:**
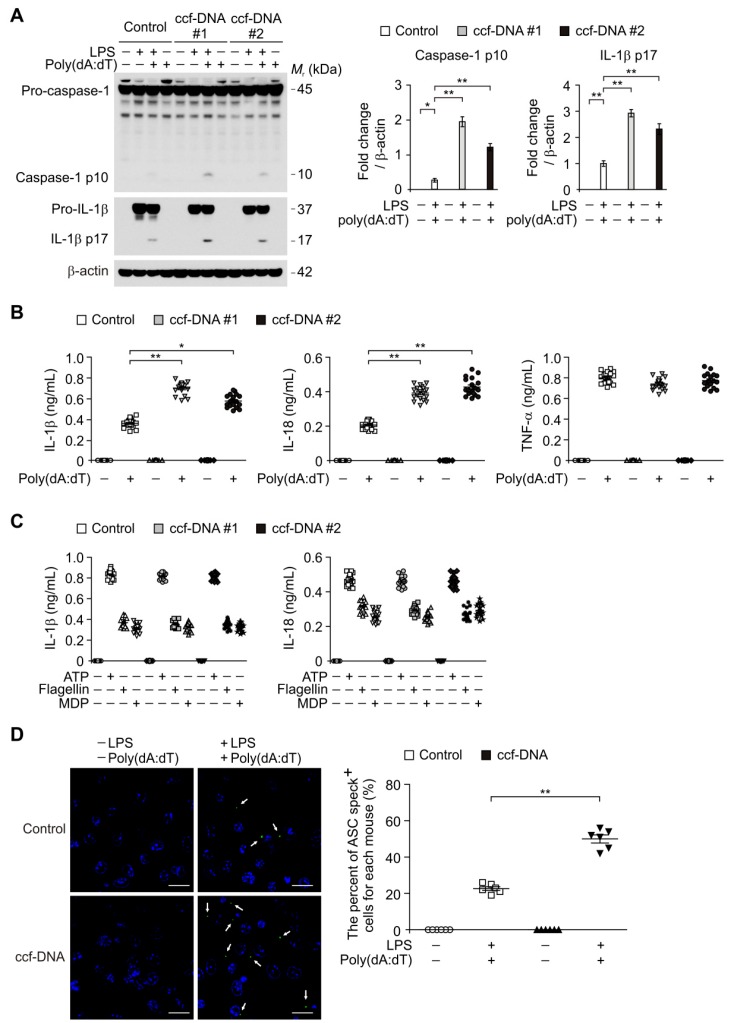
The mtDNA from patients with type 2 diabetes induced absent in melanoma 2 (AIM2) inflammasome activation in macrophages. (**A**) Representative immunoblot analysis for caspase-1 and IL-1β (left), and densitometry quantification of caspase-1 p10 and IL-1β p17 levels (normalized to levels of β-actin) (right) from LPS-primed WT BMDMs were stimulated with poly(dA:dT) and ccf-DNA (ccf-DNA #1 and ccf-DNA #2), or with poly(dA:dT) and vehicle control (control). For immunoblots, β-actin was used as loading control. (**B**) Quantification of IL-1β (left), IL-18 (middle) and TNF-α (right) secretion from LPS-primed WT BMDMs were stimulated with poly(dA:dT) and ccf-DNA (ccf-DNA #1 and ccf-DNA #2), or with poly(dA:dT) and vehicle control (control). (*n* = 18 mice per group). (**C**) Quantification of IL-1β and IL-18 secretion from LPS-primed WT BMDMs were stimulated with ATP or flagellin or MDP, and ccf-DNA (ccf-DNA #1 and ccf-DNA #2) or vehicle control (control). (*n* = 18 mice per group). (**D**) Representative immunofluorescence images (total 100 cells in 15 individual images per group) (left) and quantification (right) of caspase recruitment domain (ASC) speck formation (white arrows) (the percent of ASC speck positive cells for each mouse) in LPS-primed WT BMDMs were stimulated with poly(dA:dT) and ccf-DNA, or with poly(dA:dT) and vehicle control (control). (*n* = 6 mice per group). Scale bars, 20 μm. Data are mean ± SD ** *p* < 0.01, * *p* < 0.05; by two-tailed *t*-test or ANOVA.

**Table 1 cells-08-00328-t001:** Baseline characteristics of non-diabetic control and patients with type 2 diabetes.

	Control	Patients with Type 2 Diabetes	*p*-Value
	(*n* = 22)	(*n* = 141)	
Age, years	51.6 ± 6.0	56.5 ± 10.7	0.003
Sex, male (%)	11 (50.0)	60 (42.6)	0.672
ccf-mtDNA, copy number (1 × 10^3^)/μL	0.1 ± 0.01	1.91 ± 0.17	<0.01
IL-1β, pg/μL	30.27 ± 0.12	38.53 ± 0.71	<0.001
HbA1c (%)	0 (0.0)	8.1 ± 1.99	1.000
Hypertension, present	0 (0.0)	103 (75.2)	1.000
BMI, kg/m^2^	23.0 ± 2.6	23.5 ± 3.5	0.464
Systolic BP, mmHg	119.1 ± 15.1	124.0 ± 13.6	0.129
Diastolic BP, mmHg	74.1 ± 8.3	77.3 ± 9.4	0.124
Hemoglobin, g/dL	14.7 ± 1.7	13.5 ± 1.7	0.001
Urea nitrogen, mg/dL	14.2 (11.5–16.4)	15.9 (12.8–20.4)	0.039
Creatinine, mg/dL	0.83 ± 0.14	0.81 ± 0.20	0.580
Albumin, g/dL	4.49 ± 0.28	4.43 ± 0.42	0.522
AST, IU/L	19.5 (18.0, 23.8)	19.0 (16.0, 25.0)	0.459
ALT, IU/L	16.5 (13.0, 23.0)	20.0 (16.0, 27.3)	0.044
Total bilirubin, mg/dL	1.09 ± 0.38	0.65 ± 0.26	<0.001
Total cholesterol, mg/dL	195.0 ± 34.5	178.7 ± 37.6	0.059
Triglycerides, mg/dL	120.5 (60.0, 141.0)	142.0 (91.5, 216.5)	0.020
Uric acid, mg/dL	5.02 ± 0.88	4.76 ± 1.33	0.367
Calcium, mg/dL	9.42 ± 0.45	9.35 ± 0.43	0.465
Phosphorus, mg/dL	3.45 ± 0.58	3.69 ± 0.58	0.077
eGFR, mL/min/1.73 m^2^	101.2 (89.7, 110.2)	99.2 (87.6, 109.9)	0.690
C-reactive protein, mg/L	0.04 (0.03, 0.08)	0.07 (0.03, 0.15)	0.193

Data are presented as mean ± SD, median (interquartile range), or count (%) as appropriate. P-values are calculated using the Student’s *t*-test for normally distributed continuous variables, the Mann–Whitney U test for non-normally distributed continuous variables, and the Pearson’s Chi-squared test for categorical variables. BMI, body mass index; BP, blood pressure; AST, aspartate aminotransferase; ALT, alanine aminotransferase; eGFR, estimated glomerular filtration rate.

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
