# Peer review of "Circulating Cell-Free mtDNA Contributes to AIM2 Inflammasome-Mediated Chronic Inflammation in Patients with Type 2 Diabetes"

_cells, 2019, doi:10.3390/cells8040328_

Round 1

Reviewer 1 Report

In this study, Dr. Bae and colleagues aim to demonstrate that circulating cell free mtDNA (ccf-mtDNA) levels are associated with chronic inflammation in patients with Type 2 Diabetes. In this regard, they report a statistically significant correlation between ccf-mtDNA copy number and HbA1c level, as well as between ccf-mtDNA copy number and IL-1beta level in patients with Type 2 Diabetes. Despite the tests showing a significant p-value, the practical importance of the discovery it outweighed by the r-values, which are 0.1728 and 0.1107; respectively. In order to assess practical importance, the correlation coefficient should be square to get R2, the “coefficient of determination”. If we do this for the r-values that the authors report, we obtain 0.029 and 0.012, respectively, which means that in the best scenario one of the variables would account for only 2.9% of the variance for the other; thus demonstrating that the correlation does not bear any actual importance. In light of this consideration, this reviewer would exclude any valuable correlations between ccf-mtDNA copy number and HbA1c and IL-1beta levels in patients with Type 2 Diabetes.

Author Response

Response to Cells Reviewer 1 Comments

Point 1: In this study, Dr. Bae and colleagues aim to demonstrate that circulating cell free mtDNA (ccf-mtDNA) levels are associated with chronic inflammation in patients with Type 2 Diabetes. In this regard, they report a statistically significant correlation between ccf-mtDNA copy number and HbA1c level, as well as between ccf-mtDNA copy number and IL-1beta level in patients with Type 2 Diabetes. Despite the tests showing a significant p-value, the practical importance of the discovery it outweighed by the r-values, which are 0.1728 and 0.1107; respectively. In order to assess practical importance, the correlation coefficient should be square to get R2, the “coefficient of determination”. If we do this for the r-values that the authors report, we obtain 0.029 and 0.012, respectively, which means that in the best scenario one of the variables would account for only 2.9% of the variance for the other; thus demonstrating that the correlation does not bear any actual importance. In light of this consideration, this reviewer would exclude any valuable correlations between ccf-mtDNA copy number and HbA1c and IL-1beta levels in patients with Type 2 Diabetes.

Response 1:

According to the Reviewer’s suggestion, we have excluded Figure 2B and 3B, which were included correlations between ccf-nDNA copy number and HbA1c levels or IL-1β levels in patients with type 2 diabetes in main figure. We have performed the analysis using Spearman correlation coefficient test in between ccf-mtDNA copy number and HbA1c levels or IL-1β levels (see new Supplementary Fig. S1 and Supplementary Fig. S3). We have provided a better interpretation of the analysis using Spearman correlation coefficient test.

According to the Reviewer’s suggestion, we have excluded figures including correlations between ccf-nDNA copy number and HbA1c levels in patients with type 2 diabetes. (see new Supplementary Fig. S2).

To reflect these new results, the following text has been added to the Results section:

Results, page 5 line 187. There was weak correlation between the elevated mtDNA levels and HbA1c levels in plasma from patients with type 2 diabetes (Supplementary Fig. S1).

Results, page 5 line 203. There was weak correlation between the elevated mtDNA levels and IL-1β levels in plasma from patients with type 2 diabetes (Supplementary Fig. S3).

Reviewer 2 Report

The paper by Bae et al investigates circulating cell-free mtDNA in type-2 diabetic patients and the relationship between quantity of mtDNA and inflammasome activation. with a sound scientific rational I recommend this paper for publication with only one minor comment to address. The authors should expand on their discussion to more thoroughly place their findings within the context of what is known in the field. 

Author Response

Response to Cells Reviewer 2 Comments

Point 1: The paper by Bae et al investigates circulating cell-free mtDNA in type-2 diabetic patients and the relationship between quantity of mtDNA and inflammasome activation. with a sound scientific rational I recommend this paper for publication with only one minor comment to address. The authors should expand on their discussion to more thoroughly place their findings within the context of what is known in the field.

Response 1:

According to the Reviewer’s suggestion, we provided the expanded discussion for our finding related to type 2 diabetes. Furthermore, we discussed the interpretation of correlation between ccf-mtDNA and HbA1c levels in the Discussion section.  

The following text has been added to the Discussion section:

Discussion, page 10.  Mitochondrial dysfunction is associated with insulin resistance and type 2 diabetes [32]. Dysfunctional mitochondria can trigger release of mtDNA into extracellular space during cellular injury and death. Although recent studies have shown that ccf-mtDNA levels were increased and linked to human metabolic diseases related to type 2 diabetes, the role of ccf-mtDNA is not well understood. Our findings provide a molecular mechanism by which ccf-mtDNA promotes chronic inflammation via AIM2 inflammasome activation in patients with type 2 diabetes. Consistent with our findings, ccf-mtDNA is linked to chronic inflammation in patients with hemodialysis and cardiovascular diseases [33,34].

Our findings suggest that the elevated mtDNA in plasma might be an important signaling molecule to induce AIM2 inflammasome-mediated chronic inflammation in type 2 diabetes. Because the mtDNA in extracellular space could be brought into the immune cells such as macrophages [35], the mtDNA might be a critical molecule for chronic inflammation in patients with type 2 diabetes.

While we demonstrated that the elevated mtDNA levels in plasma from patients with type 2 diabetes contribute to inflammatory response in macrophages, the elevated ccf-mtDNA might have a role in regulating cellular signaling and function in type 2 diabetes. The ccf-mtDNA could be associated with the metabolic dysfunction by either direct or indirect mechanism in various organs, such as skeletal muscle and liver, during type 2 diabetes. Previous studies have shown that the exogenous mitochondrial DAMPs contribute to hyperglycemia and insulin resistance [36,37]. Consistent with previous studies, our findings suggest the elevated mtDNA is associated with HbA1c levels in plasma from patients with type 2 diabetes.

Although our results showed that the interaction between the elevated mtDNA and HbA1c levels, the correlation of these two factors was weak. These results suggest that other factors could be involved in dysregulation of glucose homeostasis in patients with type 2 diabetes. Thus, the roles of circulating mtDNA in impaired glucose homeostasis in type 2 diabetes may be clarified by further studies.

Reviewer 3 Report

General comments

The Authors reported that the ccf-mtDNA levels are associated with chronic inflammation in patients with type 2 diabetes. The mtDNA levels were elevated and associated with hemoglobin A1c (HbA1c) levels in plasma from patients with type 2 diabetes compared to healthy subjects. The elevated mtDNA levels were positively correlated with interleukin-1β (IL-1β) levels in patients with type 2 diabetes. Furthermore, the mtDNA from patients with type 2 diabetes induced AIM2 inflammasome-dependent caspase-1 activation and IL-1β and IL-18 secretion in macrophages.

Overall, the paper is clear and scientifically sound, and the reported results are of potential interest; however, the methodological approach is weak and should be strengthened.

Specific Comments

I suggest improving the paper as follows:

Page 6: In figure 1 it has been reported the ccf-DNA levels increase in plasma from patients with type 2 diabetes comparing ssDNA and dsDNA concentrations from the plasma of 22 healthy subjects (control) and 141 patients with type 2 diabetes (T2D). I think that the comparison of 141 patients versus 22 controls could be misleading; therefore I strongly suggest to calculate the effect size of these comparisons. Unlike significance tests, effect size is independent of sample size, statistical significance, on the other hand, depends upon both sample size and effect size.

Page 6: Figure 2B shows the correlation analysis between the mtDNA levels and the HbA1c levels in plasma from patients with type 2 diabetes. From the dots distribution reported in this figure, It appears that the variables could have an asymmetric distribution, in this condition the Pearson’s correlation analysis could be not the best choice, I suggest to repeat this analysis using Spearman correlation test which is distribution-independent.

Page 7: figure 3B: I have the same concern reported for figure 2b.

Page 5 line 206: The Authors interestingly show that the mtDNA from patients with type 2 diabetes induced AIM2 inflammasome activation in macrophages and release of IL-1β and IL-18. I think that these data should be further supported by performing additional experiments to confirm the role of AIM2. For example, I suggest repeating the experiments reported in figure 4 in the presence of specific inhibitors for caspase-1 and/or NLRP3.

Author Response

Response to Cells Reviewer 3 Comments

Point 1: Page 6: In figure 1 it has been reported the ccf-DNA levels increase in plasma from patients with type 2 diabetes comparing ssDNA and dsDNA concentrations from the plasma of 22 healthy subjects (control) and 141 patients with type 2 diabetes (T2D). I think that the comparison of 141 patients versus 22 controls could be misleading; therefore I strongly suggest to calculate the effect size of these comparisons. Unlike significance tests, effect size is independent of sample size, statistical significance, on the other hand, depends upon both sample size and effect size.

Response 1:

According to the Reviewer’s suggestion, we have performed the analysis for the effect size of comparisons about ssDNA and dsDNA concentrations from the plasma of 22 healthy subjects (control) and 141 patients with type 2 diabetes (T2D). We have provided Cohen's d effect size d value d = 1.84 for comparing ssDNA and d = 0.69 for comparing dsDNA, respectively (see new Figure 1A and 1B legend).

To reflect these new results, the following text has been added to the Figure legends section:

Figure legends, page 6 line 243. Cohens d effect size d=1.84.

Figure legends, page 6 line 245. Cohens d effect size d=0.69.

Point 2: Page 6: Figure 2B shows the correlation analysis between the mtDNA levels and the HbA1c levels in plasma from patients with type 2 diabetes. From the dots distribution reported in this figure, It appears that the variables could have an asymmetric distribution, in this condition the Pearson’s correlation analysis could be not the best choice, I suggest to repeat this analysis using Spearman correlation test which is distribution-independent.

Response 2:

According to the Reviewer’s suggestion, we have performed the analysis using Spearman correlation coefficient test in between ccf-mtDNA copy number and HbA1c levels (see new Supplementary Fig. S1). We have provided a better interpretation of the analysis using Spearman correlation coefficient test.

To reflect these new results, the following text has been added to the Results section:

Results, page 5 line 187. There was weak correlation between the elevated mtDNA levels and HbA1c levels in plasma from patients with type 2 diabetes (Supplementary Fig. S1).

Point 3: Page 7: figure 3B: I have the same concern reported for figure 2b.

Response 3:

According to the Reviewer’s suggestion, we have performed the analysis using Spearman correlation coefficient test in between ccf-mtDNA copy number and IL-1β levels (see new Supplementary Fig. S3). We have provided a better interpretation of the analysis using Spearman correlation coefficient test.

To reflect these new results, the following text has been added to the Results section:

Results, page 5 line 203. There was weak correlation between the elevated mtDNA levels and IL-1β levels in plasma from patients with type 2 diabetes (Supplementary Fig. S3).

Point 4: Page 5 line 206: The Authors interestingly show that the mtDNA from patients with type 2 diabetes induced AIM2 inflammasome activation in macrophages and release of IL-1β and IL-18. I think that these data should be further supported by performing additional experiments to confirm the role of AIM2. For example, I suggest repeating the experiments reported in figure 4 in the presence of specific inhibitors for caspase-1 and/or NLRP3.

Response 4:

According to the Reviewer’s suggestion, we validated the role of mtDNA-induced AIM2 inflammasome activation on caspase-1-dependent IL-1β and IL-18 secretion. We have performed new experiments to examine whether mtDNA-induced AIM2 inflammasome activation is critical for capase-1-dependent IL-1β and IL-18 secretion. We found that Z-VAD, a specific caspase-1 inhibitor, suppressed caspase-1 activation, IL-1β cleavage and secretion of IL-1β and IL-18 in response to mtDNA and poly(dA:dT) in LPS-primed BMDM compared to vehicle control (see new Supplementary Fig. S4). These results support that mtDNA-induced AIM2 inflammasome activation is critical for capase-1-dependent IL-1β and IL-18 secretion.

To reflect these new results, the following text has been added to the Results section:

Results, page 2 line 222. Moreover, we examined whether mtDNA-induced AIM2 inflammasome activation could regulate caspase-1 activation-dependent IL-1β and IL-18 secretion. We analyzed caspase-1 activation and IL-1β and IL-18 secretion in BMDMs pre-treated with Z-VAD, a selective caspase-1 inhibitor, before poly(dA:dT) and ccf-DNA stimulation after LPS incubation. Z-VAD suppressed caspase-1 activation, IL-1β cleavage and secretion of IL-1β and IL-18 in response to ccf-DNA and poly(dA:dT) stimulation relative to vehicle control, while TNF-α was unchanged (Supplemental Figure S4A, S4B).

According to the Reviewer’s suggestion, we have performed a new experiment to further characterize the role of mtDNA in AIM2 inflammasome activation. We examined whether mtDNA could affect NLRC4 or NLRP1 inflammasome activation in BMDMs. We found that mtDNA had no effect on the secretion of IL-1β and IL-18 in response to muramyldipeptide (MDP), a NLRP1 inflammasome activator, or flagellin, a NLRC4 inflammasome activator in LPS-primed BMDMs (see new Figure 4C). These results support that mtDNA promotes AIM2 inflammasome activation.

To reflect these new results, the following text has been added to the Results section:

Results, page 2 line 228. In contrast, the ccf-DNA (ccf-DNA #1 and ccf-DNA #2) had no effect on the secretion of IL-1β and IL-18 in response to flagellin, a NLRC4 inflammasome activator, or muramyldipeptide (MDP), a NLRP1 inflammasome activator (Figure 4C).

Round 2

Reviewer 3 Report

I thank the authors for their willingness to improve the paper

Author Response

Response to Cells Reviewer 3 Comments

Point 1: I thank the authors for their willingness to improve the paper.

Response 1:

I appreciate all your comments to improve our paper. Thank you so much.
